# Protein-crystal detection with a compact multimodal multiphoton microscope

Qing-di Cheng[1,7], Hsiang-Yu Chung[2,3,7], Robin Schubert [1,4], Shih-Hsuan Chia[2,3], Sven Falke [1,5], Celestin Nzanzu Mudogo [1], Franz X. Kärtner[2,3,5 ✉], Guoqing Chang[2,6 ✉] & Christian Betzel[1,5 ✉]

There is an increasing demand for rapid, effective methods to identify and detect protein micro- and nano-crystal suspensions for serial diffraction data collection at X-ray free-electron lasers or high-intensity micro-focus synchrotron radiation sources. Here, we demonstrate a compact multimodal, multiphoton microscope, driven by a fiber-based ultrafast laser, enabling excitation wavelengths at 775 nm and 1300 nm for nonlinear optical imaging, which simultaneously records second-harmonic generation, third-harmonic generation and three-photon excited ultraviolet fluorescence to identify and detect protein crystals with high sensitivity. The instrument serves as a valuable and important tool supporting sample scoring and sample optimization in biomolecular crystallography, which we hope will increase the capabilities and productivity of serial diffraction data collection in the future.

[1] Laboratory for Structural Biology of Infection and Inflammation, University of Hamburg, c/o DESY, Building 22a Notkestrasse 85, 22607 Hamburg, Germany. [2] Center for Free-Electron Laser Science, DESY, Notkestrasse 85, 22607 Hamburg, Germany. [3] Physics Department, University of Hamburg, Luruper Chaussee 149, 22761 Hamburg, Germany. [4] XFEL Biological Infrastructure Laboratory at the European XFEL, Holzkoppel 4, 22869 Schenefeld, Germany. [5] The Hamburg Centre for Ultrafast Imaging, University of Hamburg, Luruper Chaussee 149, 22761 Hamburg, Germany. [6] Beijing National Laboratory for Condensed Matter Physics, Institute of Physics, Chinese Academy of Sciences, Beijing 100190, China. [7]These authors contributed equally: Qing-di Cheng, Hsiang-Yu Chung. ✉email: franz.kaertner@cfel.de; guoqing.chang@iphy.ac.cn; christian.betzel@uni-hamburg.de

Sequential diffraction data collection at high-intensity micro-focus synchrotron beam lines and at X-ray free-electron lasers has become a routine method for serial synchrotron X-ray crystallography and serial femtosecond crystallography[1–4]. It offers new opportunities in structural biology, e.g., allowing data collection at room temperature and to perform time-resolved experiments[5–7]. The size and dimensions of protein crystals required for diffractive imaging at high-brilliance micro-focus X-ray sources is continuously decreasing, down to a few micrometers or even sub-micrometer regime[8,9]. In this context, the demand to produce, detect, and, in particular, to monitor and image crystal suspensions upon preparation and just prior to data collection has substantially increased[10]. As bright-field microscopy has certain limitations in detection of mostly colorless biomolecular crystals and limitations to differentiate between salt and protein crystals, as protein microcrystal samples may contain salt crystals as well, due to commonly high concentration of precipitants applied within crystallization experiments, new imaging concepts emerged utilizing various physical principles. For example, second-order nonlinear imaging of chiral crystals (SONICC) employs second-harmonic generation (SHG) to analyze crystallization experiments. Commercial imaging instruments are particularly designed to analyze crystalline suspensions in multi-well plates[11]. However, those imaging systems utilizing SHG are restricted to a rather limited detection sensitivity for protein crystals grown in higher-symmetry space groups. For some diffraction data collection experiments, in vivo-grown protein crystals are used as well[12,13]. The situation to identify and image such crystals is similar but even more complex, as in vivo crystals are tiny and mostly appear as small needles and crystal suspensions containing in addition some remaining cell debris, even after intensive purification.

Therefore, novel and highly sensitive techniques and instruments are required not only to detect small crystals grown in vitro or in vivo, but also capable to distinguish between salt and protein crystals. To achieve this goal, we designed and developed an innovative multiphoton microscope (MPM) driven by a multi-color fiber-based ultrafast laser source. By frequency doubling the output of an ultrafast Er-doped fiber laser (EDFL), centered at 1550 nm wavelength, the resulting 775 nm pulses enable the simultaneous excitation of SHG from non-centrosymmetric crystals and of ultraviolet (UV) fluorescence through three-photon absorption of intrinsic fluorophores of aromatic amino acids (AAAs) in proteins. Using nonlinear fiber-optic approaches for wavelength conversion, femtosecond pulses at 1300 nm wavelength are simultaneously generated, allowing another modality, third-harmonic generation (THG) nonlinear microscopy. The measurements in this work show and confirm that our in-house developed MPM instrument can reliably detect micro-sized in vivo and in vitro protein crystals and can distinguish protein from salt crystals.

## Results

### Detection of in vitro-grown protein crystals utilizing the MPM system

To analyze the feasibility and detection sensitivity of the multimodal MPM system, we utilized different protein crystals grown in space groups with high symmetry. Lysozyme crystals with dimensions of ~35 μm and tetragonal symmetry (space group: P4$_3$2$_1$2), shown in Fig. 1a, no SHG signal was detected applying a commercial imaging system, but the two-photon UV signal can be detected to identify the big lysozyme crystal. In addition, unlike crystals with large dimensions, smaller-sized protein crystals with higher symmetry are expected to be even more difficult to detect using SHG, due to false-negative signals resulting from their limited volume. In Fig. 1b–e, micro-sized

protein crystals of lysozyme, proteinase K (space group: P4$_3$2$_1$2), thaumatin (space group: P4$_1$2$_1$2), and thermolysin (space group: P6$_1$22) are shown, which are difficult to be identified applying bright-field microscopy, and respectively imaged applying a commercially available SONICC instrument. Only large crystals showed two-photon excited UV fluorescence (2PEUVF) and no SHG signal could be detected. To highlight features and advantages of the MPM instrument in comparison to a commercial SHG imaging system nano-sized tetragonal crystals were prepared and applied for comparative imaging experiments.

In Fig. 1f–j, the imaging results for milli-sized lysozyme, micro-sized lysozyme, thaumatin, proteinase K, and thermolysin crystals, which all grow in a higher-symmetry space group, are shown, highlighting that our MPM instrument exhibits high detection sensitivity even for crystals grown in higher-symmetry space groups. Figure 1b taken by the commercial imaging system shows that 2PEUVF reveals only bigger crystals and no SHG is detected, whereas smaller lysozyme crystals are visible using three-photon excited UV fluorescence (3PEUVF) and SHG modalities of the home-built MPM instrument, as shown in Fig. 1g.

THG images of a suspension of micro-sized thaumatin crystals mixed with larger sodium-tartrate crystals are shown in Fig. 1k. The figure shows that the imaging modality can provide information about sample interfaces and therefore allows to image the shape of both thaumatin and sodium-tartrate crystals. Although Fig. 1k shows that a considerable SHG signal can be detected from orthorhombic sodium-tartrate crystals and from tetragonal thaumatin crystals, 3PEUVF signals are only detected for protein crystals. This demonstrates that a differentiation between salt and protein crystals is possible, as long as the protein contains some AAAs.

A variety of different protein crystals with different space groups were selected for the imaging experiments, as listed in Supplementary Table 1. Supplementary Fig. 1a–d show the results of SHG and UV imaging experiments for glucose isomerase, bovine serum albumin, insulin, and lactamase applying the in-house developed MPM imaging system. Further, as shown in Supplementary Fig. 1a, b, for bovine serum albumin and insulin crystals, the SHG and THG signals do not fully overlay with each other, whereas SHG signals overlay with 3PEUVF signals. This indicates that the samples contain protein crystals (as SHG and 3PEUVF are positive) as well as amorphous particles (THG positive and SHG negative). These results clearly demonstrated that micrometer-sized crystals of lysozyme, thaumatin, proteinase K, and thermolysin, which are more difficult to be detected applying only SHG, can be efficiently detected using the MPM instrument. Also, the MPM imaging system reduces the risk of false-negative results for crystal detection by providing high signal sensitivity and a tunable polarization of the excitation.

### Polarization dependence of SHG

On the other hand, we also encountered false-negative situations: small protein crystals hardly generate any SHG, which might lead to the situation that protein crystals can be missed. Previous research has shown that the choice of laser polarization can strongly influence the SHG signal intensity, leading to a large modulation of the SHG signal strength even for small-sized protein crystals[14]. Therefore, angular dependence of SHG polarization can be used to confirm SHG and analyze the optical anisotropy of the sample. Such a polarization anisotropy can be used to determine the orientation of the sample related to the polarization of the excitation light[15]. We representatively examined the polarization dependence of SHG in nano-sized thaumatin crystals. The linearly polarized excitation beam was changed from 0° to 90° in steps of 10° by

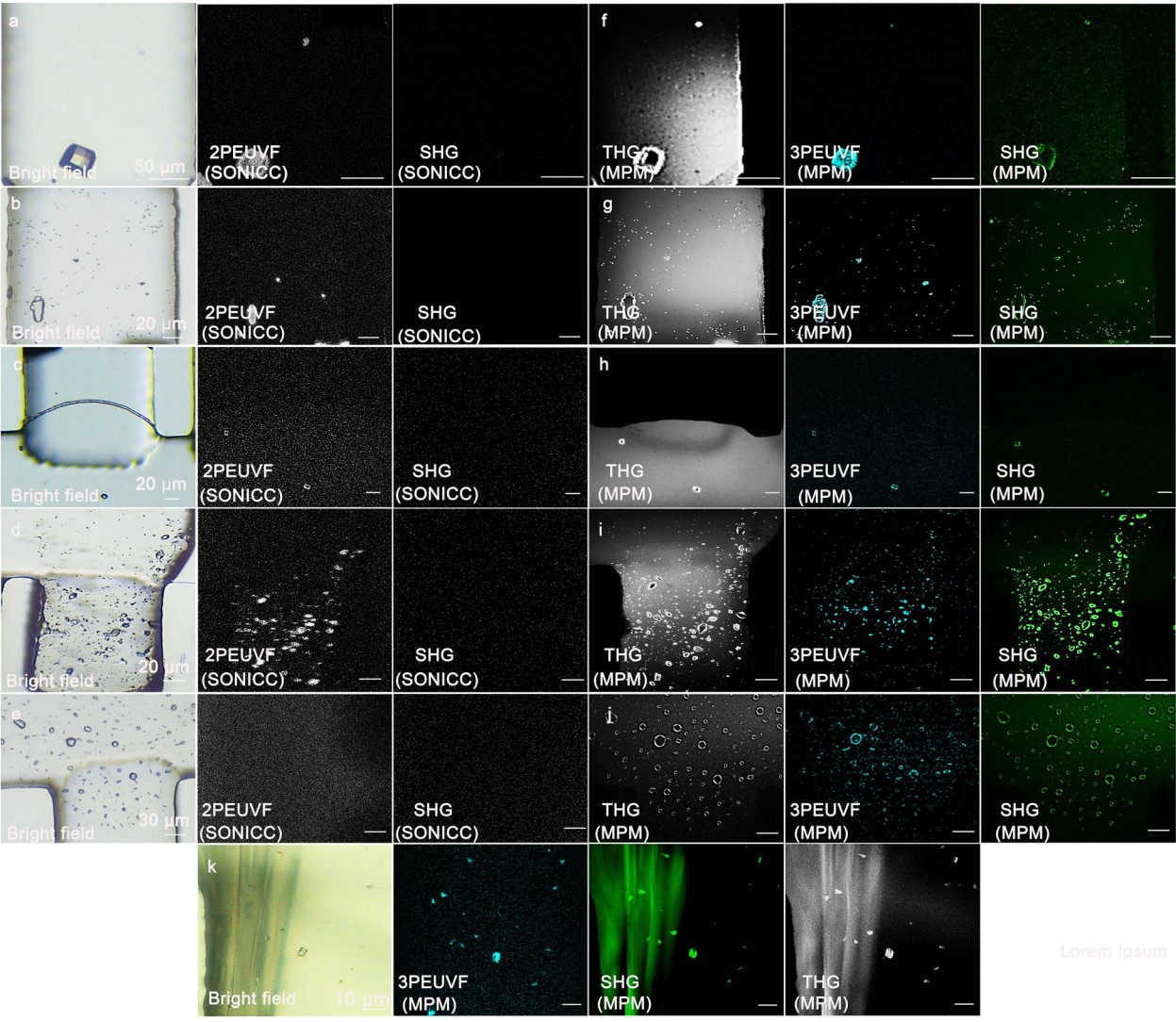

**Fig. 1 Images of different in vitro protein crystals selected for the imaging experiments.** The method of detection and imaging is indicated in the lower left of each image. **a–e** SONICC imaging of a large lysozyme crystal, micro-sized lysozyme crystals, micro-sized proteinase K crystals, micro-sized thaumatin crystals, and micro-sized thermolysin crystals. **f–j** Multimodal multiphoton imaging methods for lysozyme crystals, micro-sized lysozyme crystals, micro-sized proteinase K crystals, micro-sized thaumatin crystals, and micro-sized thermolysin crystals. **k** Multimodal multiphoton imaging methods for micro-sized thaumatin crystals and sodium-tartrate crystals.

rotating a half-wave plate (HWP) and the resulting images are shown in Supplementary Movie 1 and Supplementary Fig. 2. The SHG signal intensity varied between its maximum and minimum and has a period of ~45° rotation, which is in good agreement with crystals of space group $(4 / m)$[16,17]. By rotating the HWP, it is also possible to optimize the SHG signal intensities obtained from crystals, so that the signal intensity no longer depends on the sample container orientation.

**Detection of in vivo-grown protein crystals applying the MPM system**. To further highlight the potential of the MPM system, we applied the system to image in vivo-grown protein crystal in cells. For the experiments, we prepared in vivo-grown crystals of PAK4 (p21 serine/threonine kinase 4), PAK4-GFP (PAK4 in complex with green fluorescent protein), and IMPDH (Inosine-5′-mono-phosphate dehydrogenase). Imaging was performed without purifying the crystals from cells. Imaging results are presented in Fig. 2a–c applying SHG, THG, and 3PEUVF, respectively,

showing the needle-like shape of the samples (see also Supplementary Fig. 3).

**SHG imaging of salt crystals**. It is well known that also salt crystals can produce SHG signals, as already shown for sodium-tartrate (Fig. 1k). Therefore, we applied for comparative imaging experiments selected salt crystals with centro-symmetry ($Li_2SO_4$ and $KH_2PO_4$ are non-centrosymmetric) and we could show that applying the MPM system even centrosymmetric salt crystals can produce SHG and THG (NaCl, KCl), which was not reported before (details for salt crystals applied for the experiments, as listed in Supplementary Table 2)[18]. We conclude that the observed SHG from the centrosymmetric salt crystals is generated at the surface of crystals, due to the symmetry breaking at the surface of the crystal, confirming the high sensitivity of the MPM system[19]. In summary, the in-house MPM instrument exhibits a particularly high sensitivity, allowing SHG, THG, and 3PEUVF detection in parallel of micro-sized and even sub-micrometer-sized protein crystals,

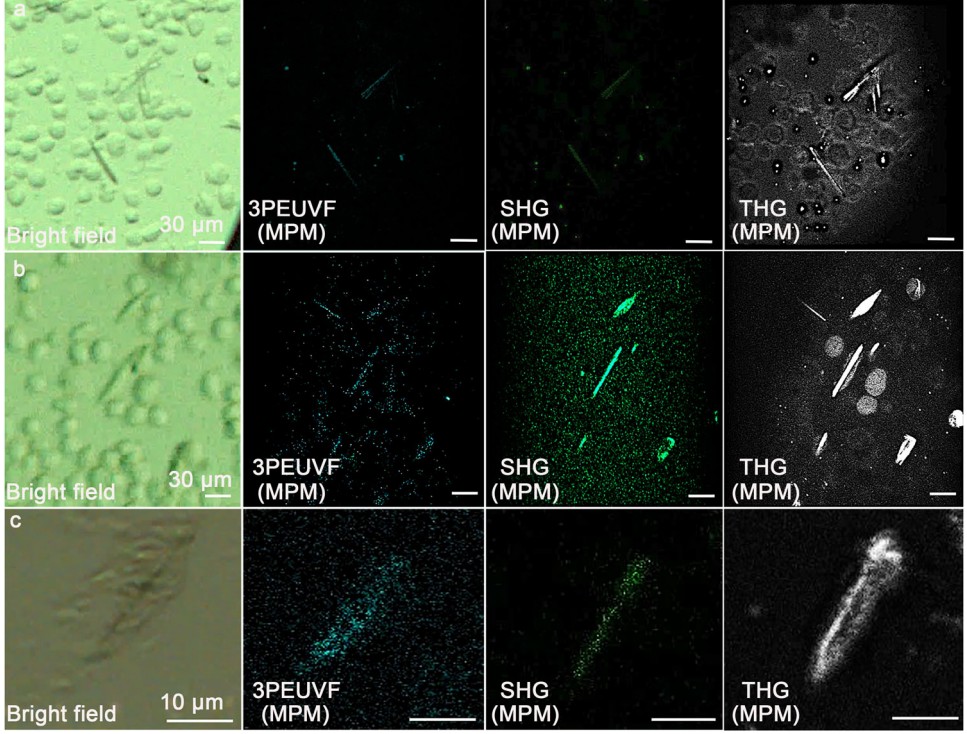

**Fig. 2 Images of PAK4 and IMPDH in vivo protein crystals selected for the imaging experiments.** The method of detection and imaging is indicated in the lower left of each image. **a** In vivo-grown protein crystal of PAK4. **b** In vivo-grown protein crystal of PAK4-GFP. **c** In vivo-grown protein crystals of IMPDH.

| Imaging methods | Characteristics and Applications | Advantages of detecting protein crystals | Limitations of detecting protein crystals | Response from different imaging methods |
|---|---|---|---|---|
| Bright-field | Sample illumination is transmitted white light, and contrast in the sample is caused by attenuation of the transmitted light in dense areas of the sample. 1. It is useful for samples that have an intrinsic color. 2. Living cells can be seen with bright-field microscopes. | Bigger-sized protein crystals can be easily detected; | 1. Nano- or micro-sized protein crystal samples that are naturally colorless and transparent cannot be seen well due to low resolution; 2. Furthermore, it cannot distinguish protein crystals and salt crystals | Samples in Fig. 1 (a)-(k) and Fig. S1 (a)-(d) can be detected (including protein crystals, salt crystals, protein amorphous materials). |
| THG | THG detects interfaces and optical heterogeneities. 1. Imaging morphogenesis in small animal models; imaging lipids in cells and tissues, as well as crystals; 2. THG can also be found at interfaces between water and large protein aggregates, such as collagen bundles or muscle fibers. | 1. Label-free; 2. Non-invasive; 3. THG does not require a specific asymmetry of the structure to be imaged; 4. Bigger-sized protein crystals and nano- or micro-sized protein crystals can be easily detected. | 1. It cannot distinguish protein crystals and salt crystals; 2. False positive from phase separation and protein amorphous materials | Samples in Fig. 1 (a)-(k) and Fig. S1 (a)-(d) can be detected (including protein crystals, salt crystals, protein amorphous materials). |
| SHG | SHG detects a material with a noncentreosymmetric structure. 1. Visualization of biological materials contain non-centrosymmetric units – including coiled-coil structures and polymeric proteins; 2. Investigation of striated muscle and fibrillar collagen, as well as crystals. | 1. Label-free; 2. Non-invasive. | It cannot distinguish protein crystals and salt crystals with a noncentreosymmetric structure; | Protein crystals in Fig. 1 (a)-(k) and Fig. S1 (a)-(d) can be detected. |
| UV fluorescence | Samples are illuminated with UV fluorescence and the fluorescence generated from aromatic amino acids like tryptophan tyrosine and phenylalanine are detected. 1. Fluorescent proteins can aid in localizing proteins, observing protein binding; 2. Protein crystals detection. | 1. Label-free; 2. Non-invasive. | False positive from phase separation. and protein aggregation. | Protein crystals and protein amorphous materials in Fig. 1 (a)-(k) and Fig. S1 (a)-(d) can be detected. |

**Fig. 3 Comparisons of different imaging methods.** Comparison of different imaging methods, summarising advantages and limitations of complementary imaging methods implemented in the MPM system.

reducing substantially the risk of false-negative results in comparison to commercially available systems. Further, the system is highly versatile with respect to changes in the optical configuration and, in this context, most flexible to utilize different sample containers, such as standard hanging- or sitting-drop crystallization plates, polydimethylsiloxane (PDMS) microfluidic crystallization chips, lipidic cubic phase plates, or capillaries in use for counter diffusion experiments.

## Discussion

The characteristics of various imaging methods for identifying protein crystals are introduced, as well as their applications

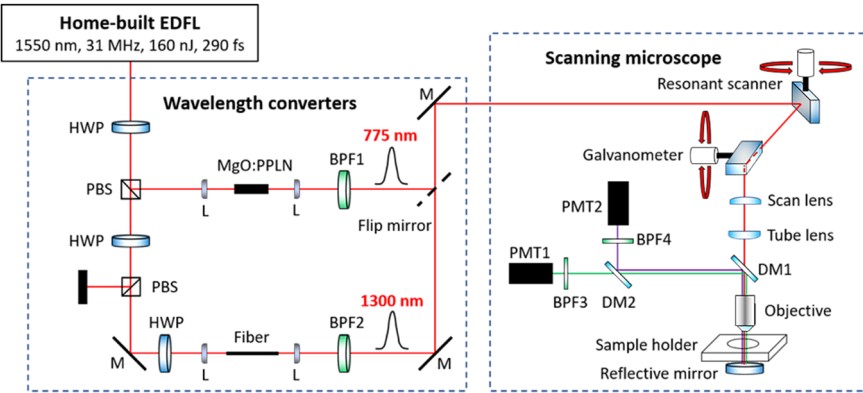

**Fig. 4 Schematic setup of the in-house constructed MPM instrument.** Layout of the Er-fiber laser system. BPF: bandpass filter, DM: dichroic mirror, HWP: half-wave plate, L: lens, M: mirror, MgO:PPLN: magnesium-doped periodically poled lithium niobate, PBS: polarization beamsplitter, PMT: photomultiplier tube. THG and 3PEUVF images are not acquired simultaneously. SHG and THG images are acquired first applying 1300 nm excitation, then SHG and UV images are acquired simultaneously by 775 nm excitation.

(Fig. 3). Combining SHG, THG, and 3PEUVF together, the system provides unique and complementary information about a crystalline sample suspension (complementary information of imaging methods are shown in the Fig. 3 and Supplementary Table 3).

The detection process of the MPM system for protein crystals is similar to the detection process of the commercial imaging system; both of them possess SHG imaging and UV imaging, the only difference is the MPM system is applying the THG imaging instead of bright-field imaging. The imaging contrast of THG originates from optical inhomogeneity[20], which serves to locate the sample.

A comparison between the results in Fig. 1a–e and those in Fig. 1f–j clearly show that the SHG signals of lysozyme crystals (space group: $P4_32_12$), thaumatin crystals (space group: $P4_12_12$), and thermolysin crystals (space group: $P6_122$) cannot be detected by the SONICC instrument but can be detected by the in-house MPM system. A comparison of parameters affecting the signal generation between the two systems is listed in Supplementary Table 4.

## Methods

**Microscopy.** SHG is detectable in non-centrosymmetric structures. Salt crystals possessing non-centro-symmetry grow from the precipitant solution during protein crystallization can also contribute to false-positive signals. To further distinguish protein crystals from salt crystals one can employ fluorescence imaging. Intrinsic AAAs—such as tryptophan, tyrosine, or phenylalanine, which are present in most proteins—emit fluorescence at ~350 nm wavelength, as a result of 280 nm excitation through one-photon absorption[21]. Simultaneously, it is found that the wavelength between 300–400 nm with the emission spectrum and the absorption properties of tryptophan produce a clear, detectable UV fluorescence signal[21]. Due to interactions between amino acids tightly packed into the crystal lattice, protein crystals can have a substantial amount of longer wavelength fluorescence (400–600 nm) when excited at 405 nm[22]. Also, one-photon UV excitation might induce photochemical damage[23]. Background suppression due to the overlap of the excitation and the emission spectra needs to be considered, which can be circumvented by multiphoton excitation. For example, 2PEUVF is an alternative to fluorescence imaging. It uses ~500 nm excitation wavelength, which has a considerable wavelength difference between excitation and emission to avoid spectral overlap.

Imaging systems were developed with options to distinguish between protein and salt crystals. These systems utilize mostly beside standard high-resolution bright-field microscopy UV excitation; further, some imaging systems utilize multimodal imaging, such as SHG and multiphoton fluorescence, which are driven by ultrafast laser sources[24,25]. For example, Nd:YAG lasers or Yb-doped fiber lasers generating pulses at ~1064 nm serve as a two-color source for multimodal imaging. The fundamental output at 1064 nm is implemented for the SHG modality. In this context, the 1064 nm pulses are frequency doubled to 532 nm in nonlinear crystals for 2PEUVF. It is noteworthy that under ~800 nm excitation, 3PEUVF of AAAs is also feasible. The corresponding SHG (~400 nm) suffers from less optical attenuation and can be effectively detected by photomultiplier tubes (PMTs), compared with the 532 nm excitation (resulting SHG at 266 nm). Typically, 800 nm

pulses are generated from ultrafast Ti:sapphire lasers. However, such solid-state lasers are bulky, sensitive to environmental fluctuations, and require accurate beam alignment and water cooling, which have spurred an intensive development of reliable sources in fiber format with compactness, robustness, and cost-effectiveness.

**The MPM system.** The MPM system consists of an in-house built EDFL, two nonlinear wavelength converters, and a scanning microscope. The EDFL was built using the well-known chirped pulse amplification scheme. It operates at 31 MHz repetition rate and generates 290 fs pulses centered at the wavelength of 1550 nm with up to 160 nJ pulse energy (corresponding to 5 W output power) after dechirping[26]. To achieve wavelength tunability, we introduced two nonlinear wavelength converters[27]. The first one is performing frequency doubling in a magnesium-doped periodically poled lithium niobate crystal and generating pulses centered at 775 nm, enabling SHG and 3PEUVF imaging. The other one employs self-phase modulation (SPM) in an optical fiber, which can substantially broaden a narrowband input spectrum to >400 nm bandwidth with well-isolated spectral lobes. For example, using an ultrafast EDFL centered at 1550 nm wavelength as the pump source, the SPM-dominated nonlinear effect can typically extend the spectrum from 1300 to 1700 nm in optical fibers within only a few cm long[26–29]. The pulse energies for 1300 nm and 775 nm are ~10 nJ (corresponding to 300 mW average power) after wavelength conversion. The excitation power after the objective is up to 60 mW for both wavelengths (corresponding to ~2 nJ pulse energy). The pulse duration for the 1300 nm pulses is ~95 fs after the objective. For 775 nm, the pulse duration after the objective is ~200 fs. Using optical filters to select the leftmost spectral lobe, we achieve nearly transform-limited pulses at 1300 nm, enabling SHG and THG imaging. We dub this methodology SPM-enabled spectral selection[26–31]. A flip mirror is introduced, allowing to choose different excitation beams (775 nm/1300 nm) entering the scanning microscope (MPM-2PKIT, Thorlabs), which consists of a resonant scanner (8 kHz) and a mirror galvanometer (up to 30 Hz) to sweep the field of view. The ×25 objective (XLPLN25XWMP2, Olympus) is water immersive with 1.05 numerical aperture and 2 mm working distance. A dichroic mirror (DM1) (F665-Di02-25 × 36, Semrock) separates the excitation beam and the emitted signal epi-collected by the objective. We implemented two filter cubes (dichroic mirror DM2, bandpass filters BPF3 and BPF4) for quick exchange between different excitation wavelength. Under 1300 nm excitation for SHG/THG detection, we use DM (FF562-Di03-25 × 36, Semrock) as DM2; a 650/50 BPF (center wavelength/full width at half maximum, unit in nm) and a 425/50 BPF from Edmund Optics) as BPF3 and BPF4. Under 775 nm excitation for SHG/3PEUVF detection, we use DM (Di01-R355-25 × 36, Semrock) as DM2; a 387/15 BPF and a 355/46 BPF (from Semrock) as BPF3 and BPF4. The PMT1 and PMT2 (H7422P-40 and H10721-210, Hamamatsu) have maximum detection sensitivity at 580 nm and 420 nm, respectively. The scheme of the MPM system is shown in Fig. 4. The acquisitions of SHG and THG are simultaneous for 1300 nm excitation. Under 775 nm excitation, SHG and 3PEUVF are simultaneously acquired as well. A software suite (ThorImageLS, Thorlabs) and a data acquisition card (PCIe-6321, National Instruments) are used to acquire images. It takes a few seconds (<10 s) to capture one image with 1024 × 1024 pixels, including the data acquisition and processing.

**Sample containers.** To allow imaging in and with a variety of commercially available crystallization plates and sample containers a particular and flexible detection platform was implemented. The setup allows visual and automatic inspection, as well as dielectrophoretic sorting, etc.[32,33]. Further, special sample containers can be coupled to the setup, which can be also utilized for real-time

investigations to follow the crystallization process of biomolecules[34,35]. Fortunately, there are very little limitations for different types of sample containers, which can be applied, owing to a flexible platform of the MPM system. Even real-time detection of samples in flow (in capillary or PMDS microfluidic chips) can be performed in future. Supplementary Table 5 shows the information about sample containers applied for the experiments. The flexible MPM imaging platform allows to use different crystallization containers, including Swissci 96-well 2-drop crystallization plates, PDMS microfluidic chips, and counter diffusion capillaries.

**Production of PDMS microfluidic chips**. For part of the experiments PDMS microfluidic chips were utilized to store micro-sized protein crystals for the imaging experiments. They consist of a mixture of silicone base with 10% (w/w) curing agent (Sylgard® 184, Dow Corning). The PDMS was thoroughly mixed (Thinky ARE-250) at 2000r/min for 2 min. Then we wrapped a 90 mm petri dish with aluminum foil and carefully manipulated the foil all around the dish, ensuring that the bottom of the foil was perfectly flat. After this, the silicon wafer (designed by Michael Heymann)[36] was positioned into a petri dish and the PDMS was carefully poured into the wafer, which was ready to be used. Later, we placed the PDMS mix and wafer in a vacuum chamber to remove bubbles and then placed them into an oven, heated at 70 °C for ~2 h. Notably, we can peel the PMDS layer of foil from the master mold when the PDMS is hard. The inputs and outputs of the microfluidic device were punched with a puncher (UniCore, Harrison), allowing the injection of protein crystal suspensions. The PDMS with micro-channels and the glass slide were simultaneously cleaned and activated with 0.4 mbar oxygen plasma (Zepto, Diener electronic) for 30 s. Then, the microfluidic channels in the PDMS structure were carefully oriented in a parallel position to the edges of the glass slide and both were shortly pressed together for bonding by a metal weight for 15 min. Finally, the PMDS was heated along with a glass slide in a 70 °C oven for 1 h to introduce tight binding.

**Preparation of in vitro-grown protein crystals**. Micro-sized and nano-sized protein crystals with different space groups were prepared by applying the batch crystallization method. Thaumatin, Proteinase K, Thermolysin, and Thaumatin were obtained from Sigma-Aldrich. A lactamase was expressed and purified in-house. All proteins were dissolved in the appropriate buffer solutions. The solutions were centrifuged at 8000 r.p.m. for 10 min and filtered through a 0.22 μm pore size filter to obtain pure and homogeneous protein solutions applied for the crystallization experiments. For batch crystallization 50 μl of protein solution and an equal volume of 50 μl precipitant were mixed and injected into the channels of the PDMS chips. To prevent evaporation wax was used to seal the access holes of the chips. All crystallization conditions are summarized in Supplementary Table 4. Prior to imaging experiments crystal suspensions were stored at room temperature (295 K).

**Preparation of in vivo-grown protein crystals**. To obtain in vivo-grown crystals, we chose insect cells (Sf9 cells) and mammalian cells (Human embryonic kidney 293 cells). Sf9 insect cells were seeded (0.5 Mio cells/ml) in serum-free medium (EX Cell® 420, Sigma-Aldrich) + 10% Penicillin–Streptomycin (Sigma-Aldrich) and infected with three corresponding P3 baculovirus constructs. The three constructs were PAK4, PAK4-GFP, and IMPDH. Infected cells were cultured/incubated at 27 °C without additional $CO_2$ for 72–96 h and cell growth was monitored by light microscopy. Construct-dependent cells were collected 4–7 days after infection and were inactivated by a mild fixation with freshly prepared 4% paraformaldehyde (PFA) (Sigma-Aldrich). This step was followed by several phosphate-buffered saline (PBS) washing and centrifugation steps to remove unbound PFA. PFA-fixed insect cells were stored in 1 × PBS (Sigma-Aldrich) at 4 °C until usage.

Mammalian cells were seeded (0.5 Mio cells/ml) in Dulbecco's modified Eagle medium (Sigma) + 10% Penicillin–Streptomycin (Sigma-Aldrich) + 5% fetal calf serum (Sigma-Aldrich) and were transfected with different Plasmids (PAK4, PAK4-GFP) via polyethyleneimine. Afterward, the transfected cells were cultured/incubated at 37 °C with 5% $CO_2$ for several days and the crystal growing was monitored by light microscopy. In mammalian cells, tiny crystals appeared after a few hours. Cells were collected 3–4 days after transfection and were inactivated by a mild fixation with freshly prepared 4% PFA, followed by several PBS washing and centrifugation steps to remove unbound PFA; the cells were stored in 1× PBS at 4 °C until usage.

**Statistics and reproducibility**. SHG signal intensities at polarization angles ranging from 0° to 90° in steps of 10° were calculated based on randomly selected particles with similar dimensions. For these calculations, a densiometric quantification was performed using the program ImageJ, available at https://imagej.net/Welcome. Particles, i.e., four in each image, shown in Supplementary Fig. 2a–j, were analyzed individually by integration of the total gray-scale pixel intensities over the definite particle areas. The obtained arithmetic mean of four intensity values for each image (Supplementary Fig. 2a–j) was plotted in context of the polarization angle, as shown in Supplementary Fig. 2k.

The reproducibility of protein crystal preparation performed in terms of the experiments was verified performing throughout the crystallization experiments

always three independent preparation batches in parallel, before collecting quantitative multiphoton microscopy data.

**Reporting summary**. Further information on research design is available in the Nature Research Reporting Summary linked to this article.

## Data availability
All data and images supporting the results of this study are available from the corresponding authors upon request.

## Code availability
To operate the MPM microscope, we used the software suite (ThorImageLS, Thorlabs), which is an open-source image acquisition program that controls Thorlab's microscope setups. The software is available via https://www.thorlabs.com/newgrouppage9.cfm?objectgroup_id=9072. Further, we used the software ImageJ to prepare and format images. The software is available at https://imagej.net/Welcome.

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

## Acknowledgements

We acknowledge financial support by the Cluster of Excellence "Advanced Imaging of Matter" of the Deutsche Forschungsgemeinschaft (DFG) - EXC 2056 - project ID 390715994, by the Helmholtz Excellence Network "Structure, Dynamics and Control on the Atomic Scale," and DFG project BE1443/29-1, by the German Aerospace Center (DLR) via project 50WB1422 and BMBF via projects 05K16GUA, 05K19GU4, and by the Joachim-Herz-Stiftung Hamburg via the project Infecto-Physics. Further, we acknowledge support obtained by the Helmholtz Association through the Helmholtz Young Investigator Group (VH-NG-804), Helmholtz Association through the Virtual Institute (VH-VI-419) "Dynamic Pathways in Multidimensional Landscapes," and Helmholtz-CAS Joint Research Group (HCJRG 201). We thank Michael Heymann for making and designing silicon wafer, Nadine Werner and Sydow Astrid for providing lactamase and in vivo protein crystals, and Dominik Oberthür and Henry Chapman (CFEL/DESY) for performing imaging experiments at the Formulatrix SONICC system. Open Access funding enabled and organized by Projekt DEAL.

## Author contributions

C.B., G.Q.C., and F.X.K. conceived the project. H.Y.C., S.H.C., and G.Q.C. designed and built the optical system. Q.D.C., C.N.M., and R.S. prepared samples. C.B., G.Q.C., S.F., and F.X.K. provided advice on biological samples and MPM system construction. Q.D.C., H.Y.C., C.B., G.Q.C., and F.X.K. wrote the paper with advice from all authors.

## Competing interests

The authors declare no competing interests.
