## [Peer Review File · Communications Biology]

Reviewers' comments:

Reviewer #1 (Remarks to the Author):

The authors describe the construction of a scanning microscope for second and third harmonic generation as well as multiphoton absorption microscopy. This is a nice piece of equipment.

In order to better assess the success of the several detection modes, a table that lists the success of various modalities could highlight similarities and differences between the methods with the various crystals (listed in Fig. 1 and S1) which are sometime difficult to detect for the non-expert from pictures only. Each table entry could refer to a different panel in Figs. 1 and S1 or maybe another figure. The table could also have an entry that points out why a certain crystal form was selected (in addition to that it was simply available in the lab).

Figure 1 is too big. It could easily be split into 2 different figures.

Lysozyme is a very special example. As there is no SHG for lysozyme, it can be easily detected with the other modalities. Please mention this explicitly for the general reader. Please indicate in the text which higher symmetries do not produce SHG and which do. Is there a similar relationship for THG?

Fig. 1, caption:

Caption for figure 1 a-f . Please indicate clearly in the figure captions that the left panels are produced by the commercially available Formulatrix SONICC instrument using the different instrument imaging methods, and the right panels are produced by the multimodal instrument as described here. Please indicated whether UV means 2PEUVF or 3PEUVF, and update the figure caption accordingly and specify what instrumentation was used. Explain why (d) does not produce a SHG signal, whereas in panel (i) strong SHG is shown.

Figure 1 can be split into two.

Fig. 2

The figure caption needs improvement: What are the purposes of a "resonant" scanner, and the "galvanometer". Please mention this in the figure caption. Maybe, you could also indicate typical wavelengths for example after the MgO:PPLN and the bandpass filter, or after the fiber.

General: There are a number of "Howevers", and "Till nows" and "Todays" that can be easily removed.

Reviewer #2 (Remarks to the Author):

This manuscript describes the design and evaluation of a nonlinear microscope for imaging protein crystals using 2 and 3 photon excited fluorescence and second and third harmonic generation for contrast. The method also includes the use of illumination polarization in second harmonic mode to better discriminate protein crystals from salt crystals and determine their space group. Polarization rotation of the exciting beam for the second harmonic signal makes sense and adds to the usefulness of the system. The instrument utilizes a lab-built fiber laser that provides femtosecond pulsed nonlinear excitation at 1300 and 775 nm to generate the four different contrast signals. Given the growing importance of serial crystallography and the need to efficiently detect small crystals for the beamline this is an important contribution. The illustration of use of the instrument on in vivo grown crystals is especially interesting. That said, there are a number of issues with the manuscript that need to be addressed.

1. Figure 1 compares images taken with a commercially available "SONICC" second harmonic and 2PEF crystal imaging system with the instrument described in the manuscript. More quantitative details are needed such as laser powers in the focus plane, objective NA's used for both setups (Olympus 25x/1.05 was used on the lab-built instrument, but no details of the SONICC objective are given), The SONICC images clearly have a much lower signal-to-noise ratio judging by the shot noise and therefore the comparison is not really a fair one. Although there are no crystals visualized in the SHG channel of the SONICC images, without knowing all the parameters it's impossible to judge the results. Is the SONICC system power limited and the noise is due to using a high gain on the PMT? The systems do differ some in design and wavelengths used, but it is surprising to not see any second harmonic in any of the SONICC images.

2. Second and third harmonic signals are forward directed, yet there is no transmission detector in the design. Is there a reflective surface below the samples like some systems use to help direct the nonlinear scattering signal back through the objective? Or are just extremely high intensities being used?

3. 1st paragraph of the methods section: "However, one-photon UV excitation might induce photochemical damage" – this true, but 2P excitation also induces damage in the focal plane – have the authors determined damage thresholds. This is related to comment 2 above – if you're using high laser powers to generate enough signal to image 2nd and 3rd harmonic in epi-mode, I would suspect this will cause some damage.

4. 1st paragraph of the methods section: "Background suppression need to be considered, due to the overlap of the excitation and the emission spectra, which can be circumvented by multiphoton excitation." Change "need" to "needs." Nonlinear imaging does have the advantage of using excitation wavelengths 100's of nanometers redder than the emission, but this implies that single photon excited fluorescence suffers from high background which is not true. There is some overlap between the red and blue edges, respectively, of the absorption and emission bands, but with the right filter selection this is never an issue.

5. 1st paragraph of the methods section: "To further distinguish protein crystals from salt crystals one can employ fluorescence imaging. Intrinsic AAAs, such as tryptophan, tyrosine, or phenylalanine, which are present in most proteins, emit fluorescence at ~350-nm wavelength, as a result of 280-nm excitation through one-photon absorption". Many protein crystals, due to interactions between amino acids tightly packed into the crystal lattice, have a substantial amount of longer wavelength fluorescence (400-600 nm) when excited at 405 nm (e.g. Lukk et al J Appl Crystallogr. 2016; 49(Pt 1):234-240).

6. Figure 2 needs more labeling on the diagram (for example, the wavelength of each exc. beam) and more details in the figure legend. Filter parameters should be fully specified (usually as center wavelength/FWHM – 600/50), rather "#84-786, #86-949, Edmund Optics" as in the methods section.

7. How do you know you're looking at 3PUVF and 3rd harmonic signals – a power vs signal plot could be shown in the supplementary section.

8. "Typically, 800-nm pulses are generated from ultrafast Ti:sapphire lasers. However, such solid-state lasers are bulky, sensitive to environmental fluctuations and require accurate beam alignment and water cooling" – I understand the need to justify the lab-built fiber laser used here, but modern Ti:S lasers are not quite as bad as the description provided – they are very stable and reliable lasers. And their pulses are more transformed limited and less noisy than many fiber lasers. Also - commercial single wavelength fs lasers are now available (e.g. the Coherent Axon series at 920 and 1024 nm, with other wavelengths such 800 nm on the way) - these are relatively cost effective. How does the laser built here compare in terms of pulse-to-pulse stability, reliability and cost?

Overall, this is a description of a custom instrument that addresses an important problem – how to best distinguish and determine the quality of micron sized protein crystals for crystallography.

Warren R. Zipfel
Cornell University

Reviewer #3 (Remarks to the Author):

Review of "Protein-crystal detection with a compact multimodal multiphoton microscope"

Summary:

The authors present a design for a multiphoton microscope for selective detection of protein crystals. This design combines measurements of second harmonic generation (SHG), three photon excited ultraviolet fluorescence (3PEUVF), and third harmonic generation (THG) to locate and distinguish crystalline protein from amorphous protein or non-protein crystals. For UV fluorescence and SHG, this microscope shows improved sensitivity than a commercially available model. Images from many different sample types are shown to support the general applicability of this microscope design for identification of samples that are suitable to serial x-ray diffraction experiments.

Overall impression:

The authors clearly described their multiphoton microscope design and thoroughly tested its applicability to identifying protein microcrystals, even with samples that are difficult for other multiphoton microscopes. I would like to see statistics from a larger number of samples that show the accuracy of this method, but the images that are shown demonstrate that this microscope is capable of generating high quality images.

Comments:

1. Could the authors discuss the use of THG further? Page two states that it originates from optical inhomogeneity and page 4 says that it can identify amorphous particles, but how does THG information aid in identifying protein crystals?
2. The laser system and microscope hardware are described well, but the data acquisition is not described. What software is used to acquire and construct the images? How long does each image take? Are all modes acquired simultaneously, as implied in the abstract, or is THG acquired separately from SHG and 3PEUVF, as implied by the flip mirror in figure 2?
3. How does the 1.05 NA water immersion objective compare to the commercial system? Is this a significant source of sensitivity improvement?
4. Could the pulse energy and duration of the beam after the wavelength conversion steps and/or at the sample position be given? Knowing what types of pulses are actually interacting with the sample is important for comparing with other systems.
5. Is there a reference to Micheal Heymann's PDMS mold design, or was it designed just for this project?
6. Can the authors mention the SONICC system used for comparison? I'm only aware of one SONICC system, but there are many other commercially available SHG and two photon excited fluorescence microscopes, which could cause confusion. If there is reason not to mention the specific model or company, could some specifications be listed, like laser power and wavelength and objective magnification and NA?
7. Scoring is mentioned in the abstract and introduction, but not in the body. Scoring results were not given for the images shown, nor for any other data sets.
8. Reordering the images in Figure 1 k-n so they match a-j (i.e. THG, UV, SHG) may be helpful.

Reviewer #1 (Remarks to the Author):

The authors describe the construction of a scanning microscope for second and third harmonic generation as well as multiphoton absorption microscopy. This is a nice piece of equipment.

Comment 1:

In order to better assess the success of the several detection modes, a table that lists the success of various modalities could highlight similarities and differences between the methods with the various crystals (listed in Fig. 1 and S1) which are sometime difficult to detect for the non-expert from pictures only. Each table entry could refer to a different panel in Figs. 1 and S1 or may be another figure. The table could also have an entry that points out why a certain crystal form was selected (in addition to that it was simply available in the lab).

Reply:

We thank the reviewer for the positive comment and the valuable suggestions to improve the manuscript. As suggested by the reviewer, we added in the supplementary material a table that contains applications and highlights similarities/differences between the methods applied to different crystals. In this table we also explain why we choose these protein crystals.

Comment 2:

Figure 1 is too big. It could easily be split into 2 different figures.

Reply:

We agree with the reviewer. Following the suggestion, we divided Figure 1 into two figures (i.e., now Figure 1 and Figure 2 in the revised version). Corresponding legends and captions were updated in the manuscript accordingly. In the revised version, Figure 1 shows comparative images of protein crystals acquired by the SONICC instrument and the home-built MPM system, respectively. Figure 2 shows data and images collected from the MPM system applied to *in vivo* grown protein crystals.

We changed the figure legend in the text, page 13, line 2, to “**Figure 1** Images of different *in vitro* protein crystals selected for the imaging experiments.

Furthermore, we changed the text, page 13, line 10, to “**Figure 2** Images of different *in vivo* protein crystals selected for the imaging experiments. (a) *In vivo* grown protein crystal of PAK4. (b) *In vivo* grown protein crystal of PAK4-GFP. (c) *In vivo* grown protein crystals of IMPDH”.

Comment 3:

Lysozyme is a very special example. As there is no SHG for lysozyme, it can be easily detected with the other modalities. Please mention this explicitly for the general reader. Please indicate in the text which higher symmetries do not produce SHG and which do. Is there a similar relationship for THG?

Reply:

In theory, the protein crystals we selected with non-centrosymmetric structures should be detected by bulk SHG imaging. However, our experiments showed that the SHG signals of lysozyme crystals, (space group: $P4_32_12$), thaumatin crystals (space group: $P4_12_12$), and thermolysin crystals (space group: $P6_122$) cannot be detected by the SONICC instrument with lower sensitivity, while SHG signals of these crystals with higher symmetry can be detected by the in-house developed MPM system. The comparison of detection sensitivity between the two systems is discussed in Reply to Comment 1 from Reviewer 2.

In the revised version (page 3, line 13), we provided more information by adding “as shown in Figure 1g. A comparison between the results in Figures 1a-1e and those in Figures 1f-1j clearly show that the SHG signals of lysozyme crystals (space group: $P4_32_12$), thaumatin crystals (space group: $P4_12_12$), and thermolysin crystals (space group: $P6_122$) cannot be detected by the SONICC instrument but can be detected by the in-house MPM system.

Unlike SHG that relies on optical non-centrosymmetry, THG exhibits no such dependency on the crystal space group and symmetry. Since THG is sensitive to optical inhomogeneity, it can always be observed at the surface/interface of crystals with different refractive indices when the laser beam is focused there. Therefore, despite different space groups and symmetries THG imaging has no limitation of detecting crystals and serves as a complementary modality compared with SHG from the bulk to detect protein crystals. We included one additional reference in page 11, line 25, T.Y.F. Tsang (1995) “Optical third-harmonic generation at interfaces”, *Physical Review A*, Vol. 52, 4116-4126, explaining the generation of THG in detail.

Comment 4:

Fig. 1, caption: Caption for figure 1 a-f. Please indicate clearly in the figure captions that the left panels are produced by the commercially available Formulatrix SONICC instrument using the different instrument imaging methods, and the right panels are produced by the multimodal instrument as described here. Please indicated whether UV means 2PEUVF or 3PEUVF, and update the figure caption accordingly and specify what instrumentation was used. Explain why (d) does not produce a SHG signal, whereas in panel (i) strong SHG is shown.

Reply:

Figure 1d shows images obtained from the SONICC instrument, while Fig. 1(i) shows imaging results obtained from the in-house MPM system. Our MPM system provides a higher signal sensitivity and a tuneable polarization of the excitation. This increases the SHG sensitivity and allows to detect more SHG signals. As a result, our system can detect protein crystals more efficiently compared with the SONICC instrument.

In the revised version, we changed the figure legend in the text in page 13, line 2, to: **Figure 1** | Images of different *in vitro* protein crystals selected for the imaging experiments.

In addition, we added the description on page 13, line 10, to: **Figure 2** | Images of different *in vivo* protein crystals selected for the imaging experiments. (a) *In vivo*

grown protein crystal of PAK4. **(b)** *In vivo* grown protein crystal of PAK4-GFP. **(c)** *In vivo* grown protein crystals of IMPDH”.

Comment 5:

Fig. 2. The figure caption needs improvement: What are the purposes of a “resonant” scanner, and the “galvanometer”. Please mention this in the figure caption. Maybe, you could also indicate typical wavelengths for example after the MgO:PPLN and the bandpass filter, or after the fiber.

Reply:

We thank the reviewer for this suggestion. Figure 2 in the original manuscript is now Figure 3 in the revised version. We improved this figure as follows:

We labeled the resulting wavelengths in the figure accordingly, 775 nm and 1300 nm—after the BPF of MgO:PPLN and fiber, respectively.

Furthermore, we explained the resonant scanner and mirror galvanometer in the revised version at page 7, line 14: “..., which consists of a resonant scanner (8 kHz) and a mirror galvanometer (up to 30 Hz) to sweep the field of view.”

Comment 6:

General: There are a number of “Howevers”, and “Till nows” and “Todays” that can be easily removed.

Reply:

Thanks for the advice, we removed all “till now” and “today”, and left only two “however”.

Reviewer #2 (Remarks to the Author):

This manuscript describes the design and evaluation of a nonlinear microscope for imaging protein crystals using 2 and 3 photon excited fluorescence and second and third harmonic generation for contrast. The method also includes the use of

illumination polarization in second harmonic mode to better discriminate protein crystals from salt crystals and determine their space group. Polarization rotation of the exciting beam for the second harmonic signal makes sense and adds to the usefulness of the system. The instrument utilizes a lab-built fiber laser that provides femtosecond pulsed nonlinear excitation at 1300 and 775 nm to generate the four different contrast signals. Given the growing importance of serial crystallography and the need to efficiently detect small crystals for the beamline this is an important contribution. The illustration of use of the instrument on in vivo grown crystals is especially interesting. That said, there are a number of issues with the manuscript that need to be addressed.

Comment 1:

Figure 1 compares images taken with a commercially available “SONICC” second harmonic and 2PEF crystal imaging system with the instrument described in the manuscript. More quantitative details are needed such as laser powers in the focus plane, objective NA’s used for both setups (Olympus 25x/1.05 was used on the lab-built instrument, but no details of the SONICC objective are given), The SONICC images clearly have a much lower signal-to-noise ratio judging by the shot noise and therefore the comparison is not really a fair one. Although there are no crystals visualized in the SHG channel of the SONICC images, without knowing all the parameters it’s impossible to judge the results. Is the SONICC system power limited and the noise is due to using a high gain on the PMT? The systems do differ some in design and wavelengths used, but it is surprising to not see any second harmonic in any of the SONICC images.

Reply:

Thanks for the comment. Below we provide a table summarizing parameters of the commercial SONICC system compared to those used in the in-house MPM system. The laser of the SONICC system operates with a pulse width of 200 fs and average power (after the objective lens) of approximately 100 mW to 1 W at 80 MHz repetition rate. The pulses are short to prevent sample damage associated with sample heating. The laser wavelength is halved with a nonlinear optical (NLO) crystal from 1064 nm to 532 nm. The resulting green light (532 nm) is used to image the sample. The equivalent two-photon absorption wavelength of the green light is 266 nm, which can excite tryptophan amino acids. The two-photon excited fluorescence (350 – 400 nm) is then collected and used to display a fluorescence image. Also, the maximum objective field of view (FOV) is 0.65 x 0.65 mm² and NA is 0.51.

In general, lower crystal symmetries are likely to produce stronger SHG signals compared to higher symmetries. That is the reason why we did not detect SHG signals using the commercial SONICC system, which has restricted sensitivity.

	SONICC	MPM System
Laser repetition rate	80 MHz	31 MHz
Objective NA	0.51	1.05

Pulse duration	200 fs	100-200 fs
Maximum FOV	0.65 x 0.65 mm ²	0.8x 0.8 mm ²

The in-house developed MPM system has the following advantages over sensitivity compared to the commercial SONICC system:

1. The NA of the objective lens is higher, which results in a smaller focus with higher laser intensity.
2. The pulse duration can be tuned to be shorter.
3. At the same excitation power, the pulse energy and peak power are higher due to the lower laser repetition rate.

Comment 2:

Second and third harmonic signals are forward directed, yet there is no transmission detector in the design. Is there a reflective surface below the samples like some systems use to help direct the nonlinear scattering signal back through the objective? Or are just extremely high intensities being used?

Reply:

We use relatively high intensity, within shorter pulses and low average power to avoid damaging samples. Since most samples and their holder/container are transparent, we also placed a reflective mirror under the sample basement, as shown now in figure 3, to improve the epi-detected nonlinear signal.

Comment 3:

1st paragraph of the methods section: “However, one-photon UV excitation might induce photochemical damage” – this true, but 2P excitation also induces damage in the focal plane – have the authors determined damage thresholds. This is related to comment 2 above – if you ‘re using high laser powers to generate enough signal to image 2nd and 3rd harmonic in epi-mode, I would suspect this will cause some damage.

Reply:

As we described in the reply to comment 2, the in-house developed MPM system has a mirror below the sample basement to reflect SHG and THG signals. This considerably supports the epi-detection using low power and short exposure times to avoid sample damage and increases the sensitivity.

Comment 4:

1st paragraph of the methods section: “Background suppression need to be considered, due to the overlap of the excitation and the emission spectra, which can be circumvented by multiphoton excitation.” Change “need” to “needs.” Nonlinear imaging does have the advantage of using excitation wavelengths 100’s of nanometers redder than the emission, but this implies that single photon excited fluorescence suffers from high background which is not true. There is some overlap

between the red and blue edges, respectively, of the absorption and emission bands, but with the right filter selection this is never an issue.

Reply:

Thanks for the comment. We changed “need” to “needs” (page 6, line 1).

The “background issues” for one-photon excitation can result from the excitation/emission wavelength overlap as well as the degraded axial resolution due to out-of-focus emission. Well-designed filters can solve the first issue and introducing pinholes as a spatial filter in the confocal microscope can solve the second one at the expense of increasing the system complexity. On the other hand, nonlinear processes provide an alternative. Besides 100s of redshift of excitation wavelength for nonlinear microscopy, the fact that most of the nonlinear signal originates from the objective focus makes MPM sensitive to optical intensity and therefore has an intrinsic sectioning ability.

Comment 5

1st paragraph of the methods section: “To further distinguish protein crystals from salt crystals one can employ fluorescence imaging. Intrinsic AAAs, such as tryptophan, tyrosine, or phenylalanine, which are present in most proteins, emit fluorescence at ~350-nm wavelength, as a result of 280-nm excitation through one-photon absorption”. Many protein crystals, due to interactions between amino acids tightly packed into the crystal lattice, have a substantial amount of longer wavelength fluorescence (400-600 nm) when excited at 405 nm (e.g. Lukk et al J Appl Crystallogr. 2016; 49(Pt 1):234–240).

Reply:

Thanks for the suggestion. In the revised version, we added the recommended reference (reference 21), which links also to reference 21, Dierks et al., Acta F. 2010; 66(Pt 4): 478–484.

Furthermore, we extended the corresponding text in page 5, line 20, “To further distinguish protein crystals from salt crystals one can employ fluorescence imaging. Intrinsic AAAs—such as tryptophan, tyrosine, or phenylalanine, which are present in most proteins—emit fluorescence at ~350 nm, as a result of 280-nm excitation through one-photon absorption. Simultaneously, it is found that the wavelength between 300 - 400 nm with the emission spectrum and the absorption properties of tryptophan produce a clear detectable UV fluorescence signal²¹. Due to interactions between amino acids tightly packed into the crystal lattice, protein crystals can have a substantial amount of longer wavelength fluorescence (400-600 nm) when excited at 405 nm²².”

Comment 6

Figure 2 needs more labeling on the diagram (for example, the wavelength of each exc. beam) and more details in the figure legend. Filter parameters

should be fully specified (usually as center wavelength/FWHM – 600/50), rather “#84-786, #86-949, Edmund Optics” as in the methods section.

Reply:

In the revised version, we modified the figure, (now Figure 3) by providing more information. We described the specification in the text as follows (page 7, line 18):

“We implemented two filter cubes (dichroic mirror DM2, bandpass filters BPF3 and BPF4) for quick exchange between different excitation wavelength. Under 1300-nm excitation for SHG/THG detection, we use DM (FF562-Di03-25×36, Semrock) as DM2; a 650/50 BPF (center wavelength/full width at half maximum, unit in nm) and a 425/50 BPF from Edmund Optics) as BPF3 and BPF4, respectively. Under 775-nm excitation for SHG/3PEUVF detection, we use DM (Di01-R355-25×36, Semrock) as DM2; a 387/15 BPF and a 355/46 BPF (from Semrock) as BPF3 and BPF4, respectively. The photomultiplier tubes PMT1 and PMT2 (H7422P-40, H10721-210, Hamamatsu) have maximum detection sensitivity at 580 nm and 420 nm, respectively.”

Comment 7:

How do you know you're looking at 3PEUVF and 3rd harmonic signals – a power vs signal plot could be shown in the supplementary section.

Reply:

As mentioned in page 7, line 20-28 onwards, we use proper bandpass filters to ensure that we image 3PEUVF and THG signals. A (425/50) bandpass filter is used to detect the THG utilizing 1300-nm excitation. For 3PUVF we use a (355/40) bandpass filter to detect the fluorescence emission originated from aromatic amino acids utilizing 775-nm excitation, which excludes the corresponding SHG (centered at 387 nm) and THG (centered at 258 nm).

We didn't measure the excitation power vs signal. We use BPFs to distinguish between 3PUVF, THG, and other nonlinear signals.

Comment 8:

“Typically, 800-nm pulses are generated from ultrafast Ti:sapphire lasers. However, such solid-state lasers are bulky, sensitive to environmental fluctuations and require accurate beam alignment and water cooling” – I understand the need to justify the lab-built fiber laser used here, but modern Ti:S lasers are not quite as bad as the description provided – they are very stable and reliable lasers. And their pulses are more transformed limited and less noisy than many fiber lasers. Also - commercial single wavelength fs lasers are now available (e.g. the Coherent Axon series at 920 and 1024 nm, with other wavelengths such 800 nm on the way) - these are relatively cost effective. How does the laser built here compare in terms of pulse-to-pulse stability, reliability, and cost?

Reply:

Cost-effective ultrafast light sources with versatile wavelength output are always desired by microscope users, working with different samples and dyes. For example,

920-nm source can excite GFP through two-photon absorption. However, the specific requirements on the excitation wavelength in this application for protein autofluorescence emission restricts the comparison with only Ti:S lasers. The scheme of frequency doubling an EDFL serves as an alternative to light sources near 800 nm. We didn't observe obvious signal (imaging) fluctuations due to pulse-to-pulse instabilities during the experiment. Regarding reliability and cost, it is difficult to compare a lab-built system with commercial lasers.

Reviewer #3 (Remarks to the Author):

The authors present a design for a multiphoton microscope for selective detection of protein crystals. This design combines measurements of second harmonic generation (SHG), three-photon excited ultraviolet fluorescence (3PEUVF), and third harmonic generation (THG) to locate and distinguish crystalline protein from amorphous protein or non-protein crystals. For UV fluorescence and SHG, this microscope shows improved sensitivity than a commercially available model. Images from many different samples are shown to support the general applicability of this microscope design for identification of samples that are suitable to serial x-ray diffraction experiments.

Overall impression:

The authors clearly described their multiphoton microscope design and thoroughly tested its applicability to identifying protein microcrystals, even with samples that are difficult for other multiphoton microscopes. I would like to see statistics from a larger number of samples that show the accuracy of this method, but the images that are shown demonstrate that this microscope is capable of generating high quality images.

Comment 1:

Could the authors discuss the use of THG further? Page two states that it originates from optical inhomogeneity and page 4 says that it can identify amorphous particles, but how does THG information aid in identifying protein crystals?

Reply:

THG is originating from optical inhomogeneity, therefore structural interfaces exhibiting different refractive indices can be visualized by THG. For example, THG can be found at interfaces between solvent and protein aggregates, as well as between solvent and protein crystals. In this context, crystals can be identified in the solution. Unlike SHG being rather weak in crystals of high symmetry, THG imaging is not tied to the condition of high symmetry, which can always reveal the outline/shape of the sample. Thus, our data reveal that THG imaging can be considered as an excellent complementary method to bright-field, SHG and UV imaging. We added more details in the text in page 2, line 18-19:

“Unlike SHG, THG has no dependence on the crystal symmetry and can be detected at interfaces between solvent and protein aggregates/crystals.”

We also included one additional reference according to comment 3, reviewer 1. T.Y.F. Tsang (1995) "Optical third-harmonic generation at interfaces", Physical Review A, Vol. 52, 4116-4126.

Comment 2:

The laser system and microscope hardware are described well, but the data acquisition is not described. What software is used to acquire and construct the images? How long does each image take? Are all modes acquired simultaneously, as implied in the abstract, or is THG acquired separately from SHG and 3PEUVF, as implied by the flip mirror in figure 2?

Reply:

We use a commercial scanning kit (MPM-2PKIT, Thorlabs) including a software suite (ThorImageLS, Thorlabs) and a data acquisition card (PCIe-6321, National Instruments) to acquire images, as described in page 7, line 26-28 and page 8, line 1-3:

"The acquisition of SHG and THG are simultaneous for 1300-nm excitation. Under 775-nm excitation SHG and 3PEUVF are simultaneously acquired as well". A software suite (ThorImageLS, Thorlabs) and a data acquisition card (PCIe-6321, National Instruments) are used to acquire images. It takes a few seconds (<10 s) to capture one image with 1024x1024 pixels, including the data acquisition and processing."

THG and 3PEUVF images are not acquired simultaneously. First, SHG and THG images are acquired applying 1300-nm excitation. Then SHG and 3PEUVF images are captured via 775-nm excitation.

Comment 3:

How does the 1.05 NA water immersion objective compare to the commercial system? Is this a significant source of sensitivity improvement?

Reply:

Using objectives with higher NA improves the generation of nonlinear signals. The objective we introduce in the MPM system features a coating for better near-infrared transmission. This allows a flexible available excitation power after the objective and can significantly improve the sensitivity.

Comment 4:

Could the pulse energy and duration of the beam after the wavelength conversion steps and/or at the sample position be given? Knowing what types of pulses are actually interacting with the sample is important for comparing with other systems.

Reply:

The pulse energies for 1300 nm and 775 nm are approximately 10 nJ (corresponding to 300-mW average power) after wavelength conversion. The excitation power after the objective is up to 60 mW for both wavelength (corresponding to approx. 2-nJ pulse energy). The pulse duration for 1300-nm pulses is ~95 fs after the objective.

For 775 nm, we only measured the pulse duration before the microscope, which is 190 fs. The one after the objective is estimated to be >200 fs.

We added the above information to the text in page 7, line 6:

“The pulse energies for 1300 nm and 775 nm are approximately 10 nJ (corresponding to 300-mW average power) after wavelength conversion. The excitation power after the objective is up to 60 mW for both wavelengths (corresponding to ~2-nJ pulse energy). The pulse duration for the 1300-nm pulses is ~95 fs after the objective. For 775 nm, the pulse duration after the objective is ~200 fs”.

Comment 5:

Is there a reference to Micheal Heymann’s PDMS mold design, or was it designed just for this project?

Reply:

Thanks for the comment, we included a reference describing the design and production of this sample holder. (Y. Gicquel, R. Schubert, S. Kapis, G. Bourenkov, Th. Schneider, M. Perbandt, Ch. Betzel, H.N. Chapman and M. Heyman, (2018) J Vis Exp. 24;(134). doi: 10.3791/57133.

Comment 6:

Can the authors mention the SONICC system used for comparison? I’m only aware of one SONICC system, but there are many other commercially available SHG and two photon excited fluorescence microscopes, which could cause confusion. If there is reason not to mention the specific model or company, could some specifications be listed, like laser power and wavelength and objective magnification and NA?

Reply:

Please see the reply to comment 1 of the first reviewer 1. We added a table to compare instrument parameters.

Comment 7:

Scoring is mentioned in the abstract and introduction, but not in the body. Scoring results were not given for the images shown, nor for any other data sets.

Reply:

Thanks for the comment. As scoring is not appropriate, we changed the word to “detect”, in abstract and introduction.

Comment 8:

Reordering the images in Figure 1 k-n so they match a-j (i.e. THG, UV, SHG) may be helpful.

Reply:

We divided figure 1 into figure 1 and 2 to show and discuss the recorded images more appropriately and clearly, as suggest by referee 2 as well.

REVIEWERS' COMMENTS:

Reviewer #1 (Remarks to the Author):

The authors sufficiently addressed the critical points. The article should be published.

Reviewer #2 (Remarks to the Author):

The authors have provided a thorough and careful response to each of the comments I had in the initial review and made many additions and changes. I have no further questions or concerns. This type of instrument will be an important tool for serial protein crystallography in the future and the manuscript is a solid demonstration of its usefulness and feasibility.

Reviewer #3 (Remarks to the Author):

The authors have addressed all of my comments by adding more technical details of the microscope systems used, reworking figures, and adding references.